# Transvaginal Ultrasound Accuracy in the Hydrosalpinx Diagnosis: A Systematic Review and Meta-Analysis

**DOI:** 10.3390/diagnostics13050948

**Published:** 2023-03-02

**Authors:** Aina Delgado-Morell, Mar Nieto-Tous, Cristina Andrada-Ripollés, Maria Ángela Pascual, Silvia Ajossa, Stefano Guerriero, Juan Luis Alcázar

**Affiliations:** 1Department of Obstetrics and Gynecology, Hospital de la Santa Creu i Sant Pau, 08025 Barcelona, Spain; 2Institute of Biomedical Research Sant Pau (IIB SANT PAU), 08041 Barcelona, Spain; 3Department of Obstetrics and Gynecology, Hospital Universitari i Politècnic La Fe, 46026 València, Spain; 4Department of Obstetrics and Gynecology, Hospital General Universitari de Castelló, 12004 Castelló, Spain; 5Department of Obstetrics, Gynecology and Reproduction, Hospital Universitari Dexeus, 08028 Barcelona, Spain; 6Centro Integrato di Procreazione Medicalmente Assistita (PMA) e Diagnostica Ostetrico-Ginecologica, Azienda Ospedaliero Universitaria-Policlinico Duilio Casula, 09042 Monserrato, Italy; 7Department of Obstetrics and Gynecology, Università degli Studi di Cagliari, 09043 Monserrato, Italy; 8Department of Obstetrics and Gynecology, School of Medicine, Universidad de Navarra, 31009 Pamplona, Spain

**Keywords:** hydrosalpinx, transvaginal ultrasound, diagnosis

## Abstract

Hydrosalpinx is a condition with a crucial prognostic role in reproduction, and its diagnosis by a non-invasive technique such as ultrasound is key in achieving an adequate reproductive assessment while avoiding unnecessary laparoscopies. The aim of the present systematic review and meta-analysis is to synthetize and report the current evidence on transvaginal sonography (TVS) accuracy to diagnose hydrosalpinx. Articles on the topic published between January 1990 and December 2022 were searched in five electronic databases. Data from the six selected studies, comprising 4144 adnexal masses in 3974 women, 118 of which were hydrosalpinxes, were analyzed as follows: overall, TVS had a pooled estimated sensitivity for hydrosalpinx of 84% (95% confidence interval (CI) = 76–89%), specificity of 99% (95% CI = 98–100%), positive likelihood ratio of 80.7 (95% CI = 33.7–193.0), and negative likelihood ratio of 0.16 (95% CI = 0.11–0.25) and DOR of 496 (95% CI = 178–1381). The mean prevalence of hydrosalpinx was 4%. The quality of the studies and their risk of bias were assessed using QUADAS-2, evidencing an overall acceptable quality of the selected articles. We concluded that TVS has a good specificity and sensitivity for diagnosing hydrosalpinx.

## 1. Introduction

Infertility is related to tubal disease in 35% of cases [1]. Distal and proximal occlusion of the tubes—at the fimbrial and at the cornual end—cause fluid filling of the tubes, distending them and leading to the formation of hydrosalpinx. In vitro fertilization (IVF) therapies have changed the reproductive chances in sterility cases related to tubal obstruction, but hydrosalpinx has a negative impact on IVF outcomes according to the most recently published systematic review and meta-analysis on the topic [2].

Hydrosalpinx can develop after the onset of post-surgical adhesions or after a hysterectomy, but more frequently, its occurrence is related to an antecedent of acute pelvic inflammatory disease (PID) [3]. Taipale et al. followed a cohort of patients diagnosed of acute PID for three months, and after this period, 26.7% (23/86) of them had a sonographic suspicion of hydrosalpinx [4].

The gold standard for the hydrosalpinx diagnosis is histological confirmation, albeit laparoscopic direct observation is also an accepted method [3]. Non-invasive techniques such as ultrasound have been proposed as having a role in its diagnosis in order to achieve adequate reproductive assessment while avoiding unnecessary laparoscopies.

Classically, since a fluid-filled tube is theoretically visible by transvaginal ultrasound (TVS) as a cystic mass, hydrosalpinxes could be identified sonographically. Nevertheless, in some series it has been proved that they could be missed, especially when the bowel is occupied by gas and/or fecal material or when adhesions alter the normal pelvic anatomy [5].

Different authors have described the main sonographic characteristics to accurately diagnose hydrosalpinx. Timor-Tritsch et al. described this lesion as being typically an elongated-shaped cystic mass with anechoic or hypoechoic content, convoluted and with the presence of incomplete septa due to the ballooning and doubling-up of the tube. These incomplete septa originate as triangular hyperechoic wall-protrusions into the lumen, not reaching the opposite wall [6], and they are found in up to 85–93% of chronic hydrosalpinxes [6,7].

The presence of the called “beads-on-a-string” sonographic pattern is also a specific and reliable sign of chronic hydrosalpinx—hyperechoic mural nodules measuring 2–3 mm seen on the transverse section of the fluid-filled tube, which correspond to the flattened and fibrotic remnants of the endo-salpingeal folds [5,6].

Hydrosalpinx can also be the result of a present acute salpingitis, and not just the chronic consequence of a previous PID [6]. Some sonographic signs, nonetheless, contribute to establishing a differential diagnosis among acute and chronic salpingitis: the tubal wall is thickened in acute PID but not in the chronic hydrosalpinx [3,8]; the typical “beads-on-a-string” sign is not visible in acute salpingitis, but the “cogwheel sign” is found instead [3,6].

Although some authors have analyzed the TVS accuracy in the diagnosis of PID, both in prospective cohorts [4,6,9] and in narrative or systematic reviews [3,5,8], most of these studies include or exclusively focus on cases with acute symptoms. However, despite hydrosalpinx being an entity with a crucial prognostic role in reproduction, to our knowledge there is not a meta-analysis addressing the accuracy of TVS in the diagnosis of this condition. The aim of the present systematic review and meta-analysis is to synthetize and report the current evidence on TVS accuracy to diagnose hydrosalpinx.

## 2. Materials and Methods

### 2.1. Protocol and Registration

This meta-analysis was performed following the recommendations of the PRISMA Statement (http://www.prisma-statement.org/, accessed on 24 January 2022), as well as the guidelines from the Synthesizing Evidence from Diagnostic Accuracy Tests (SEDATE) [10,11]. The protocol was not registered. To start the search process, the researchers previously defined all inclusion and exclusion criteria to select the articles. The researchers also agreed how the data extraction should be made and how the quality assessment of the studies should be conducted, as well as the criteria used for such a purpose.

Institutional Review Board approval was waived because of the study design.

### 2.2. Data Sources and Search Process

In order to find eligible studies, three of the authors (A.D.M., M.N.T., and C.A.R.) independently screened five electronic databases—Scopus, Cochrane, ClinicalTrials.gov, PubMed/Medline, and Web of Science—during January 2023, selecting the studies published between January 1990 and December 2022 that were written in English, German, French, and Spanish. 

The search terms were the following: “hydrosalpinx” or “adnexal mass” and “transvaginal” and “ultrasound”. Search strategy is shown in Appendix A.

### 2.3. Study Selection and Data Collection

The titles and abstracts selected from the five electronic databases were independently screened by the same three authors to identify the duplicated and irrelevant articles. Then, the remaining articles were read full-text to identify the ones that applied to the following three inclusion criteria:(1)Prospective and retrospective cohort studies, which include patients with a transvaginal ultrasound assessment and a suspect diagnosis of hydrosalpinx or other adnexal masses.(2)Definitive histologic diagnosis of the adnexal mass after surgical removal, including confirmative diagnosis of hydrosalpinx.(3)Reported data that would allow constructing a 2 × 2 table to estimate true positive, true negative, false positive, and false negative values for the sonographic diagnosis of hydrosalpinx.

The following exclusion criteria were stablished: studies not related to the topic under review, letters to the editor, commentaries, reviews, consensus documents, and studies where no data were available to construct a 2 × 2 table.

Additional interesting papers on the matter were searched by reading the reference list of our previously selected full-text reading papers.

The Patients, Intervention, Comparator, Outcomes, and Study Design (PICOS) criteria used for inclusion and exclusion of studies were recorded.

The diagnostic accuracy from the ultimately selected studies was retrieved independently by the same three authors. Disagreements that came up during the study selection and data extraction were solved by consensus between these same three authors or by a fourth author (J.L.A.) when the first three could not arrive at a consensus.

### 2.4. Risk of Bias in Individual Studies

The Quality Assessment of Diagnostic Accuracy Studies-2 (QUADAS-2) tool [12] was the chosen one for assessing the quality of all the papers included in the meta-analysis. The QUADAS-2 includes four domains: (1) patient selection, (2) index test, (3) reference standard, and (4) flow and timing. For each domain, the risk of bias and concerns about applicability (not applying the second one to flow and timing) were assessed and rated as low, high, or unclear risk. This quality assessment was aimed to evaluate the overall quality of the studies and their potential sources of heterogeneity.

The first three authors (A.D.M., M.N.T., and C.A.R.) evaluated the methodological quality independently. Disagreements arising during the process were solved by discussion between them and the fourth author (J.L.A.).

For the patient selection domain, the risk of bias was assessed on the study’s design (a retrospective design was considered as high risk), by considering the inclusion and exclusion criteria adequacy and whether the recruitment was consecutive/random or not. For the index test domain, it was evaluated how the transvaginal ultrasonography was performed and interpreted—when the sonographic diagnostic criteria were stated and reasonable it was considered as low risk. For the reference standard domain, it was assessed whether the pathologist was blinded for the previous image-suspected diagnosis. For the Flow and Timing, surgery >180 days after the ultrasound diagnosis was considered as high risk.

Unclear risk was stated when the corresponding information for each domain was not reported in the study.

### 2.5. Statistical Analysis

The three same authors extracted information on the diagnostic performance of transvaginal ultrasound for hydrosalpinx.

The pooled sensitivity, specificity, positive likelihood ratio (LR+), negative likelihood ratio (LR-), and diagnostic odds ratio (DOR) were estimated using a random effects model. Subsequently, we used likelihood ratios to characterize the clinical utility of a test and to estimate the post-test probability of disease [13].

Post-test probabilities were calculated considering the mean prevalence of hydrosalpinx (pre-test probability) and using the positive and negative likelihood ratios, plotted on Fagan’s nomogram.

Heterogeneity for sensitivity and specificity was studied using Cochran’s Q statistic and the I^2^ index [14]. A *p*-value < 0.1 indicated heterogeneity. I^2^ values of 25%, 50%, and 75% were considered to mean low, moderate, and high heterogeneity, respectively [14].

We plotted forest plots for sensitivity and specificity of all studies. If heterogeneity existed, meta-regression was used for assessing covariates that could explain this heterogeneity. The covariates analyzed were sample size, year of publication, and prevalence of hydrosalpinx in each article.

In order to illustrate the relationship between sensitivity and specificity, summary receiver operating characteristic (sROC) curves were plotted. Lastly, using Deeks’ Asymmetry Test publication bias was assessed [15].

All analyses were performed using MIDAS and METANDI commands in STATA version 12.0 for Windows (Stata Corporation, College Station, TX, USA). A *p*-value < 0.05 was considered statistically significant.

## 3. Results

### 3.1. Search Results

The electronic search provided 2787 citations. After excluding 1029 duplicate records, 1758 citations remained. After reading titles, 1626 citations were ruled out (papers not related to the topic and reviews). Three authors (A.D.M., M.N.T., and C.A.R.) read the abstracts of the remaining 132 papers, and 99 more citations were dropped (papers not related to the topic, reviews, case reports, letters to the editor, and commentaries). Thirty-three papers remained for full-text reading, after which 27 papers were excluded due to absence of a definitive reference standard (histopathological diagnosis) or to lack of data needed to build the 2 × 2 table to estimate true positive, true negative, false positive, and false negative values for the sonographic diagnosis of hydrosalpinx. Six papers were ultimately included in the qualitative and quantitative synthesis [16,17,18,19,20,21]. A flowchart summarizing the literature search is shown in Figure 1.

### 3.2. Characteristics of Included Studies

Six studies published between July 2000 and December 2022 were included in the final analysis [16,17,18,19,20,21]. They reported on 4144 adnexal masses in 3974 women. Among these 4144 adnexal masses, 118 cases had histopathological diagnosis of hydrosalpinx confirmed after surgery. Mean prevalence of hydrosalpinx among the selected studies was 4.0%, ranging from 0.6% to 6.8%.

All the studies recruited the patients in a prospective manner, although two of them analyzed the data retrospectively [16,20] and two others did not specify whether the recruiting was consecutive or not [17,21].

All studies reported the mean age of the included patients except for Bhatty et al., of which none reported their menopausal status [17]. Among the rest of the studies, one study included premenopausal patients only [18] and the four others [16,19,20,21] included both premenopausal and postmenopausal patients.

Bhatty et al. focused exclusively on symptomatic (presenting pelvic pain, menstrual disorders and/or dyspareunia) or infertile women with a suspicion of adnexal mass [17].

Guerriero et al. included patients who underwent surgery for infertility, pelvic pain, uterine fibroids, endometrial hyperplasia and/or adnexal masses [18]. Women with acute genital inflammation were excluded. Yazbek et al. selected symptomatic women (abdominal pain, increased abdominal girth and/or palpable abdominal mass) with a sonographic suspicion of a benign adnexal mass, excluding all tumors with malignant sonographic criteria [21]. The primary object of this study was to assess the value of preoperative ultrasound examination in predicting the feasibility of laparoscopic surgery for benign adnexal masses, while a secondary objective was to evaluate the correlation among transvaginal ultrasound, laparoscopic findings, and the definitive histopathological diagnosis.

Regarding the ultrasound examination and the diagnostic suspicion for hydrosalpinx, all the authors stating the sonographic diagnostic criteria used the pattern recognition method [16,18,19,20,21]. Only three of the cited articles specify the expertise level of the sonographers. In Alcázar’s study all the explorations were performed by two of the authors, who were expert sonographers [16]. In Sokalska’s study, 80% of the examinations were carried out by level III examiners, according to the European Federation of Ultrasound in Medicine and Biology criteria [20]. On the contrary, in Sayasneh’s study, trained but non-expert sonographers were the only operators [19]. In the latter study, 37 different sonographers performed the examinations. The sonographer was blinded for the definitive histopathological diagnosis in all the studies. In all of the studies, except one, there is a lack of information regarding whether the pathologist was blinded for the sonographic suspicion. Alcazar et al. reported that pathologists were not blinded to ultrasound findings [16].

Table 1 shows the PICOS features of all the studies included.

### 3.3. Methodological Quality of Included Studies

The QUADAS-2 assessment of the risk of bias and of the concerns regarding the applicability of the selected studies is shown graphically in Table 2.

The recruiting design was prospective in all six studies but only three of them [16,18,19] specified that it was consecutive. In two other studies this was not specified [17,20]. The last study [21] was considered as having a high risk of patient selection bias due to its selection of patients—patients with sonographically suspected malignant masses were excluded.

Regarding the bias on the index test, all the authors sufficiently specified the sonographic diagnostic criteria—low risk of bias—except for Bhatty et al., in whose article the diagnostic criteria are unclear [17].

For the reference standard domain, only one study [16] was considered of high risk due to the pathologist not being blinded to the previous sonographic suspicion in most of the cases. The rest of the studies did not provide this information and were considered as unclear.

Regarding the flow and timing, the time elapsed between the index test and reference standard was unclear in two studies [17,21]. In the remaining four studies, it was considered as low risk since all the patients were operated within less than 120 days after the index test [16,18,19,20].

Concerning applicability, all studies were considered as low risk regarding patient selection (target population: patients with suspicion of hydrosalpinx or adnexal mass), index test (ultrasound), and reference standard (histopathological diagnosis) domains.

### 3.4. Diagnostic Performance of Transvaginal Ultrasonography for Hydrosalpinx

Table 3 shows the diagnostic performance reported in each study. Overall, the transvaginal ultrasonography has a pooled estimated sensitivity for diagnosing hydrosalpinx of 84% (95% confidence interval (CI) = 76–89%) with a pooled specificity of 99% (95% CI = 98–100%), an LR+ of 80.7 (95% CI = 33.7–193.0), an LR– of 0.16 (95% CI = 0.11–0.25), and a DOR of 496 (95% CI = 178–1381).

Observed heterogeneity is low for sensitivity (I^2^ = 0.0%; Cochran Q = 0.76; *p* = 0.98) and high for specificity (I^2^ = 82.68%; Cochran Q = 28.87; *p* < 0.001). The forest plot is shown in Figure 2.

Meta-regression shows that neither sample size, year of publication nor prevalence of hydrosalpinx in each article explained the heterogeneity observed for the specificity. However, when re-analyzing the data excluding the two studies with lower positive predictive value (PPV) [17,20], the heterogeneity for the pooled estimated specificity is lower, without changes in this last value (specificity = 99%) (Figure 3).

The HSROC curve for diagnostic performance of transvaginal ultrasound for hydrosalpinx is shown in Figure 4. The area under the curve was 0.86 (95% CI = 0.83–0.89).

Fagan’s nomogram shows that the transvaginal ultrasound suspicion for a hydrosalpinx increases the pre-test probability of correctly diagnosing it from 4% to 77%, while the absence of the sonographic diagnosis of hydrosalpinx decreases the pre-test probability from 4% to 1% (Figure 5).

Publication bias was not observed according to Deeks’ Asymmetry Test (*p* = 0.97), as shown in Figure 6.

## 4. Discussion

### 4.1. Summary of Evidence

In this review and meta-analysis, we focused on TVS accuracy to diagnose hydrosalpinx. We found six studies including data from 4144 adnexal masses and among them 118 were confirmed as hydrosalpinx on histology after surgical removal. Pooled prevalence of hydrosalpinx in these six articles is 4.0%.

The pooled estimated sensitivity is 84%, with very low observed heterogeneity (I^2^ = 0.00). The pooled estimated specificity is 99%, with a remarkable heterogeneity (I^2^ = 82.68) which cannot be explained by the differences in sample size, publication year, or on prevalence among the selected studies. Notwithstanding, it could be caused by the low PPV found in two of the studies [17,20] since it decreases to I^2^ = 5.23 when excluding these articles.

The article published by Bhatty et al. is the one with the lowest specificity (94.68%) and a presents a low PPV (50.00%) as well [17]. It is also the one with the lowest quality assessment at QUADAS-2, since the authors do not provide clear information on the patient selection and methodology used. Notwithstanding, due to its small sample size (n = 100, number of hydrosalpinxes = 10), the inclusion or exclusion of this specific study in the statistical analysis does not have a major impact on the pooled results. By contrast, if this study is excluded in conjunction with the one by Sokalska et al., which includes a bigger sample size (n = 1,066, number of hydrosalpinxes = 21) but with a very low PPV (43.90%) as well [20], the heterogeneity for the pooled estimated specificity substantially decreases.

The differences in TVS performance among the included studies could be related to differences in the sonographer’s experience, but this is not clearly stated in three out of the six studies [17,18,21].

### 4.2. Interpretation of the Results and Relevance of the Topic

We do think that our study might be clinically relevant. There is increasing evidence showing that hydrosalpinx has a deleterious impact in IVF results because of its association to decreased implantation, clinical pregnancy, and ongoing pregnancy rates; and to higher risk of ectopic pregnancy and miscarriage [2]. This has been related to the negative impact of hydrosalpinx fluid on endometrial receptivity [22,23], to a toxic effect on early embryos [24], or to a mechanic or vascular alteration interfering with implantation [25,26].

Various approaches have been proposed for managing hydrosalpinx in the context of IVF. According to the last meta-analysis on this topic [2], the best IVF results are related to hydrosalpinx removal (salpingectomy) compared to other conservative managements, like proximal tubal occlusion—either surgical or hysteroscopic occlusion—or hydrosalpinx ultrasound-guided aspiration with or without sclerotherapy.

Although there are sufficient studies in the literature showing an improvement of the IVF performance after salpingectomy in the context of hydrosalpinx [27], to our knowledge no-one has yet assessed the overall evidence on the diagnostic accuracy of TVS for hydrosalpinx, even when it is the most used diagnostic tool for this entity in reproduction clinics.

Therefore, we think that our paper provides new insight on this major problem with direct clinical implications in the reproduction clinical setting. Our expectations are that our work will help clinicians interpret the findings of TVS.

According to our results, ultrasonography has a 99% pooled specificity. Therefore, after diagnosing hydrosalpinx through pattern recognition there is low probability in finding a different adnexal mass or no masses at all during its laparoscopic removal. By contrast, the overall pooled sensitivity and LR+ are lower, suggesting that some hydrosalpinxes could be missed or confused with other adnexal masses. This fact is also relevant, especially when considering the impact of the presence of hydrosalpinx in IVF outcomes. Additionally, another important point is the inter-observer reproducibility of TVS diagnosis of hydrosalpinx. As far as we know, there is no study addressing this issue. The assessment of this issue could help to define which are the more precise ultrasound features for establishing the diagnosis of hydrosalpinx, and probably to define the need for refining or re-defining the diagnostic criteria.

Overall, it is important to acknowledge that the population of all the studies included in the current meta-analysis are women with adnexal masses of all ages, not general population nor infertile women. It is well known that in infertile women, tubal disease could represent up to 35% of the causes of infertility [1] with a mean prevalence of hydrosalpinx between 10% and 13% when diagnosed by ultrasound [28]. Therefore, it could be presumed that with an increased prevalence of hydrosalpinx in this specific population, TVS could have a better performance.

### 4.3. Strengths and Limitations

The main strength of our study is that this is the first meta-analysis addressing this topic. There has been no previous meta-analysis in the literature in which the correlation between sonography and histology in the specific diagnosis of hydrosalpinx was assessed. Romosan and Valentin reported a narrative review about the ultrasound diagnosis of pelvic inflammatory disease (PID) [3]. These authors reported on seven papers that assessed the sonographic features of PID using laparoscopy and/or endometrial biopsy as reference standard. The number of patients included in these studies ranged from 18 to 77 women. Most patients had clinical suspicion of acute PDI. Therefore, only a few cases were assessed for the diagnosis of hydrosalpinx (92 patients in two studies), which is a consequence of chronic disease. These authors reported that the presence of incomplete septa was the most specific sign of hydrosalpinx (present on 85–93% of the cases).

Regarding the quality of the included articles in our meta-analysis, we consider that it is a strength that all of them collected the data prospectively, that a quite big sample size was achieved, and that histological confirmation of the diagnosis was properly registered in all the cases. In addition, we included studies that encompass a very wide range of time (1990–2022), hence covering the majority of the published evidence on hydrosalpinx, which is an entity that was already classically described during the 1990 decade.

Moreover, five out of six articles included specify the sonographic diagnostic criteria for hydrosalpinx—pattern recognition—instead of using prediction models that solely assess the benign/malignant sonographic suspicion (such as ADNEX model, O-RADS, Simple Rules, RMI, etc.), and that all of these studies clearly describe which are the TVS-patterns being used for this identification [6,18,29]. In fact, it is our opinion that diagnosis using these prediction models cannot be related to a specific diagnosis of a given adnexal mass and, therefore, could not be used for the specific diagnosis of hydrosalpinx. This is important for Reproductive Medicine units, since pattern recognition would be the unique approach for diagnosing hydrosalpinx.

On the other hand, we consider as our major limitation the fact that the number of studies included in the meta-analysis is low, being only six. Furthermore, although the total number of cases with adnexal masses can be described as considerably large, the number of suspected—and later confirmed by histology—hydrosalpinxes among the patients included in each study is limited.

### 4.4. Future Research Agenda

As it has been pointed, there are very few studies focusing on TVS accuracy for hydrosalpinx diagnosis, specifically for concrete ultrasonographic diagnostic criteria or the pattern corresponding to hydrosalpinx and not to acute salpingitis.

Moreover, as has been summarized, a considerable part of the studies focusing on hydrosalpinx diagnosis and sonographic criteria, were performed more than 10 to 20 years ago. Since there have been major technological improvements in the sonographic field since then, it could be of interest to up-date the existing evidence. Furthermore, as stated above, an assessment of inter-observer reproducibility of current criteria for ultrasound diagnosis of hydrosalpinx is needed.

In addition, prospective randomized studies in large series of patients assessing TVS performance to diagnose hydrosalpinx in the specific population of sterile women would be of major clinical relevance.

Finally, evidence regarding which are the differences in TVS accuracy depending on the examiner’s experience would be useful for clinical practice.

## 5. Conclusions

Accurate diagnosis of hydrosalpinx is a subject of major importance in an era of assisted reproduction and IVF, given its high prevalence among infertile women and its impact on IVF outcomes, while being a potentially and easily treatable condition.

Being capable of positively diagnosing it by a non-invasive technique such as TVS is key in a group of patients to whom performing diagnostic laparoscopies can have major fertility consequences.

Our review and meta-analysis prove that TVS has good accuracy with diagnosis of hydrosalpinx, with high specificity and sufficient sensitivity and, therefore, it can be considered a reliable diagnostic technique.

## Figures and Tables

**Figure 1 diagnostics-13-00948-f001:**
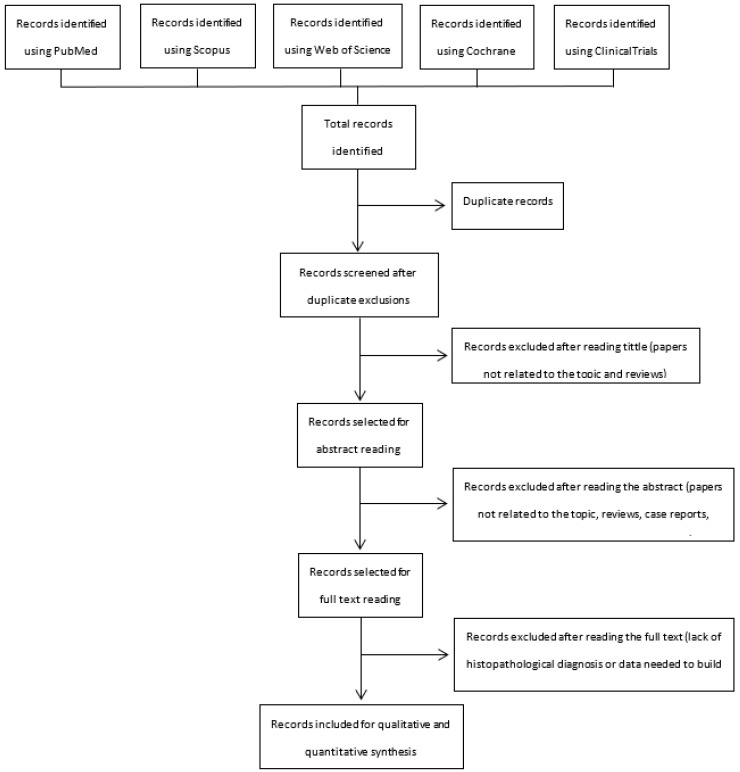
Flowchart summarizing the literature search indicating the exclusion process and the number of studies finally included in the current meta-analysis.

**Figure 2 diagnostics-13-00948-f002:**
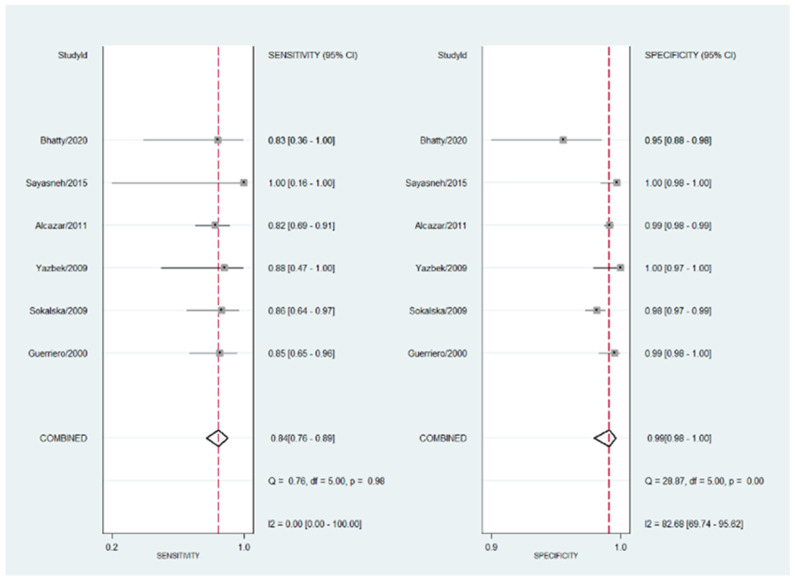
Forest plot for sensitivity and specificity for all the studies included in the meta-analysis. The heterogeneity is shown as well as the pooled sensitivity and specificity. Refs. [16,17,18,19,20,21].

**Figure 3 diagnostics-13-00948-f003:**
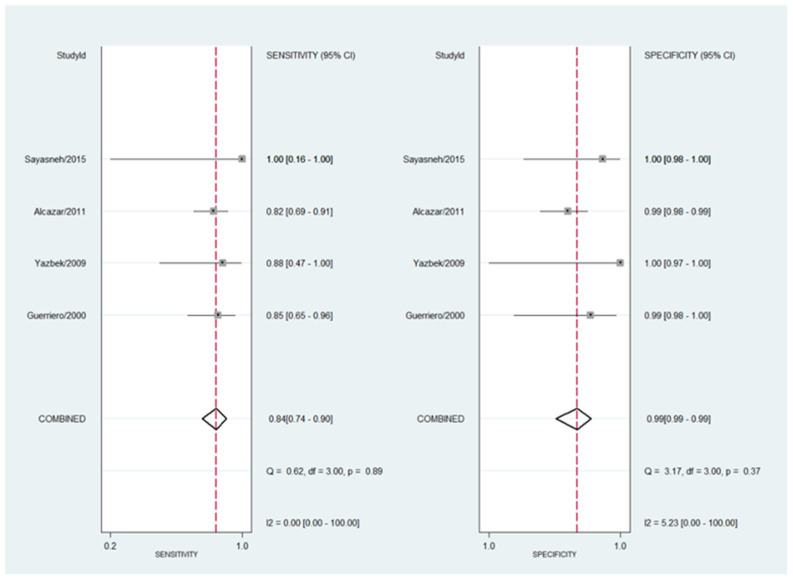
Forest plot for sensitivity and specificity for the four studies with the highest PPV included in the meta-analysis [17,20]. The heterogeneity is shown as well as the pooled estimated sensitivity and specificity. Refs. [16,18,19,21].

**Figure 4 diagnostics-13-00948-f004:**
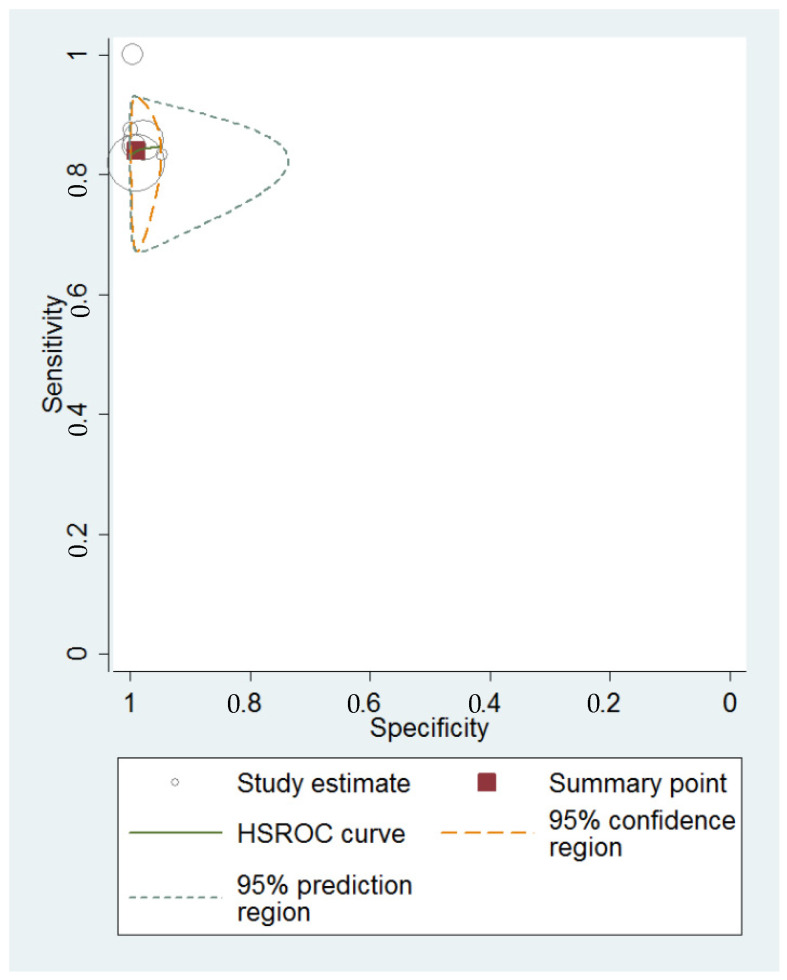
Summary HSROC curve for TVS diagnostic accuracy for hydrosalpinx, showing the sensitivity and specificity for each study and pooled estimation. The orange dashed line around the summary point estimate (red quadrate) represents the 95% confidence region. The green dotted line showing the 95% prediction contour corresponds to the predicted performance considering each of the studies.

**Figure 5 diagnostics-13-00948-f005:**
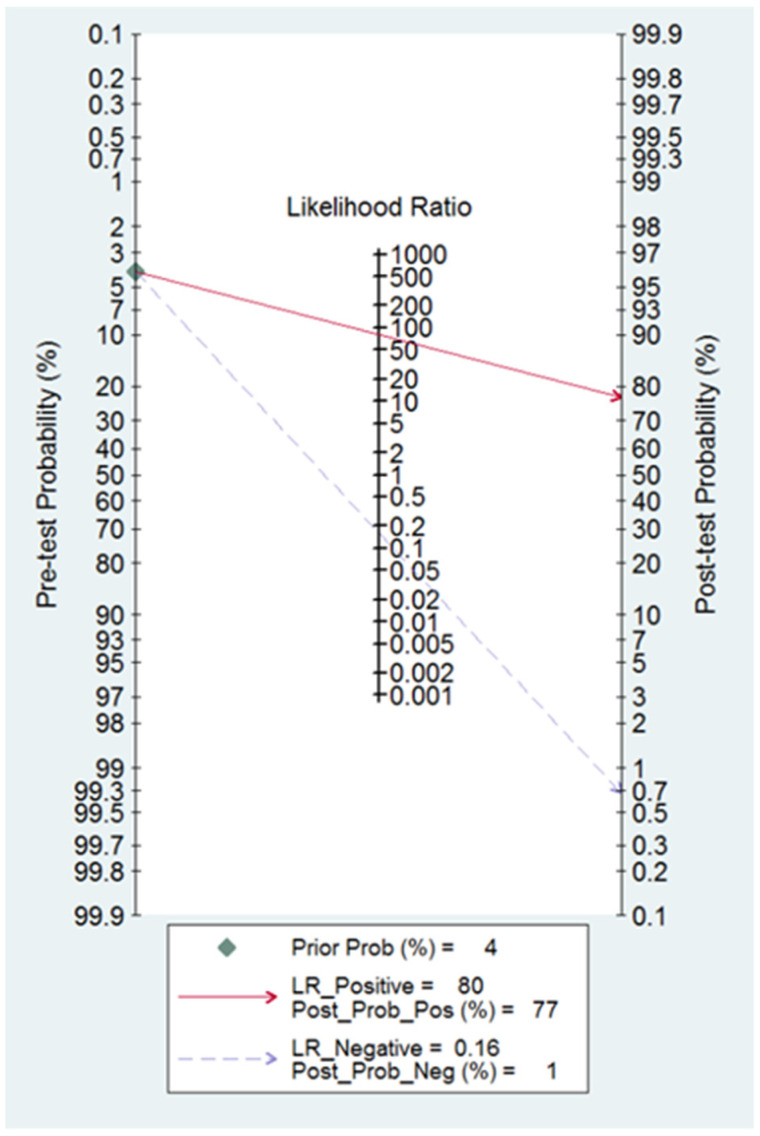
Fagan nomogram for TVS diagnostic accuracy for hydrosalpinx. This shows how depending on the positive or negative TVS diagnosis, the pre-test probability changes.

**Figure 6 diagnostics-13-00948-f006:**
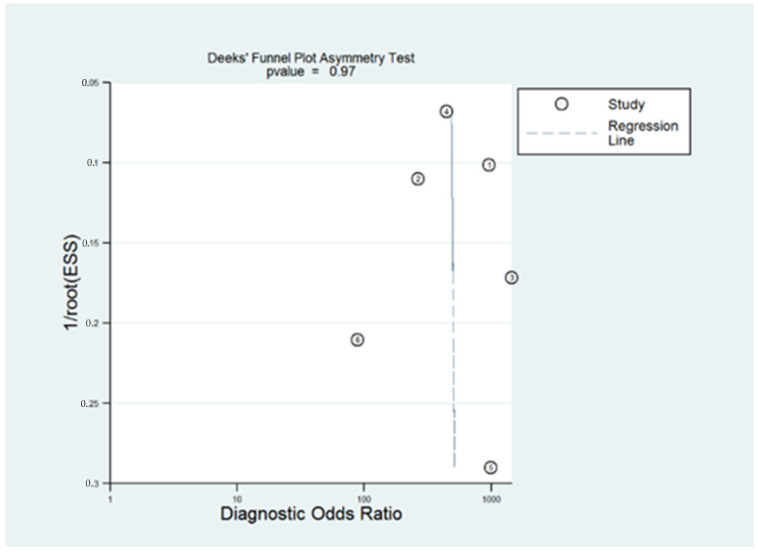
Deeks’ Funnel Plot Asymmetry Test showing the publication bias of the included articles.

**Table 1 diagnostics-13-00948-t001:** Main characteristics of the studies included in the meta-analysis (according to PICOS criteria).

Author	Year	Country	Study’s Design	N of Patients	Patients’ Mean Age	Menopausal Status	N of Masses	N of Suspected Hydrosalpinxes	N of Confirmed Hydrosalpinxes	N of Examiners	Experience of the Examiner	Blinded Examiner	Index Test	Reference Test	Time Elapsed from TVS to Surgery
Alcázar	2011	Spain	Prospective ^	1980	42.6	Pre- and postmenopausal	2146	66	55	2	Expert	yes	TVS	Histology	≤1 week
Bhatty	2020	Pakistan	Prospective	100	NA	NA	100	10	6	NA	NA	yes	TVS	Histology	NA
Guerriero	2000	Italy	Prospective	378	32.3	Premenopausal	378	24	26	NA	NA	yes	TVS	Histology	≤2 days
Sayasneh	2015	UK	Prospective	313	47	Pre- and postmenopausal	301	3	2	37	Trained	yes	TVS	Histology	≤120 days
Sokalska	2009	Poland	Prospective ^	1066	47.4	Pre- and postmenopausal	1066	41	21	NA	Expert *	yes	TVS	Histology	≤120 days
Yazbek	2007	UK	Prospective	137	35	Pre- and postmenopausal	153	7	8	NA	NA	yes	TVS	Histology	NA

N: number; NA: information not available; TVS: transvaginal sonography. ^ Retrospective analysis of prospectively collected data and recruited patients. * Only 80% of the TVS was performed by expert sonographers.

**Table 2 diagnostics-13-00948-t002:** QUADAS-2 assessment of the studies included in the meta-analysis.

Study	Risk of Bias				Concerns of Applicability		
	Patient Selection	Index Test	Reference Standard	Flow and Timing	Patient Selection	Index Test	Reference Standard
Alcázar	Low risk	Low risk	High risk	Low risk	Low concern	Low concern	Low concern
Bhatty	Unclear	Unclear	Unclear	Unclear	Low concern	Low concern	Low concern
Guerriero	Low risk	Low risk	Unclear	Low risk	Low concern	Low concern	Low concern
Sayasneh	Low risk	Low risk	Unclear	Low risk	Low concern	Low concern	Low concern
Sokalska	Unclear	Low risk	Unclear	Low risk	Low concern	Low concern	Low concern
Yazbek	High risk	Low risk	Unclear	Unclear	Low concern	Low concern	Low concern

**Table 3 diagnostics-13-00948-t003:** Diagnostic performance of each of the studies included in the meta-analysis.

Author	Sensitivity	Specificity	PPV	NPV
Alcázar	81.82%	99.00%	68.18%	99.52%
Bhatty	83.33%	94.68%	50.00%	98.89%
Guerriero	84.62%	99.43%	91.67%	98.87%
Sayasneh	100.00%	99.68%	66.67%	100.00%
Sokalska	85.71%	97.80%	43.90%	99.71%
Yazbek	87.50%	100.00%	100.00%	99.32%

PPV: Positive Predictive Value. NPV: Negative Predictive Value.

## Data Availability

Data are available upon request.

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
