# Peer review of "Transvaginal Ultrasound Accuracy in the Hydrosalpinx Diagnosis: A Systematic Review and Meta-Analysis"

_diagnostics, 2023, doi:10.3390/diagnostics13050948_

Round 1
Reviewer 1 Report
Delgado-Morell et al. present a systematic review wherein they investigated the accuracy of transvaginal ultrasound for the diagnosis of hydrosalpinx. They conclude that the evidence on current literature suggest that TVS is a sensitive and specific tool for the diagnosis of hydrosalpinx. The manuscript is well-written and informative.
Some minor comments that need to be addressed:
1. Please provide the full search strategy in the supplement
2. Please expand on the potential advantages of being able to utilize TVS for the diagnosis of hydrosalpinx and whether the current diagnostic criteria need to be further studied/modified for improving reproducibility of a TVS diagnosis of hydrosalpinx.
Author Response
Dear Reviewer
Thanks for your valuable comments.
These are our responses:
1. Please provide the full search strategy in the supplement
We have added the supplement as requested
2. Please expand on the potential advantages of being able to utilize TVS for the diagnosis of hydrosalpinx and whether the current diagnostic criteria need to be further studied/modified for improving reproducibility of a TVS diagnosis of hydrosalpinx.
We have added some sentences in the Discussion section
Sincerely yours
Juan Luis Alcazar
Reviewer 2 Report
This meta-analysis is well-done , the subject is quite common
Author Response
Dear Reviewer
Thanks for your comments. We appreciate.
Certainly the subject is a common disease. For this reason we wanted to perform this meta-analysis
No change have made in the manuscript
Sincerely yours
Juan Luis Alcazar
Reviewer 3 Report
This is an interest topic, as accurate diagnosis of hydrosalpinx is a subject of major importance in the era of ART, given its high prevalence among infertile women and its impact on IVF outcomes.
I would congratulate the authors for the first meta-analysis addressing this topic. All the article included collected the data prospectively, a significant sample size was achieved, and histological confirmation of the diagnosis was a required end-point. The wide range of time (1990-2022), covers the majority of the published evidence on hydrosalpinx, which is another strength of the study.
The main results show that ultrasound investigations has a 99% pooled specificity. Therefore, after diagnosing hydrosalpinx through pattern recognition there is low probability to find a different adnexal mass or no masses at all during surgery. Still, the overall pooled sensitivity and LR+ are lower, suggesting that some hydrosalpinxes could be missed or confused with other adnexal masses.
The paper definitely provides new insight on this condition with direct clinical implications, and helpful for the professionals involved, to interpret the findings of TVS related to adnexal findings.
Author Response
Dear Reviewer
Thanks for your comments. We appreciate them.
No change have benn made in the manuscript
Sincerely yours
Juan Luis Alcazar
Reviewer 4 Report
The manuscript is a systematic review and meta-analysis on transvaginal ultrasound (TVS) accuracy in the hydrosalpinx diagnosis.
The Introduction is well written defining the medical condition of hydrosalpinx, its causes and effects, as well as currently employed diagnosis methodologies including histological confirmation, laparoscopic observation, and transvaginal ultrasound. The authors declare that, based on their knowledge, there is no meta-analysis addressing the accuracy of TVS in the diagnosis of this condition; therefore, justifying the need for this systematic review and meta-analysis.
The Materials and Methods section clearly states that the authors followed the PRISMA and SEDATE guidelines, their inclusion/extrusion criteria for selected articles, the authors in charge of selecting the articles and their period of publication (January 1990 to December 2022), as well as the IRB waiver due to the study design.
The authors used QUADAS-2 tool to assess the quality of all the papers included in the meta-analysis, clearly explaining the four domains used: 1. Patient selection, 2. Index test, 3. Reference standard, and 4. Flow and timing; each with its respective risk of bias and concerns about applicability.
It is also clearly stated how sensitivity, specificity, positive likelihood ratio, negative likelihood ratio, and diagnostic odds ratio were estimated using random effects model. For sensitivity and specificity heterogeneity the authors used Cochran’s Q statistic and the I^2 index. They chose a p-value < 0.1 to indicate heterogeneity and the I^2 values (25%, 50%, 75%) to indicate low, moderate, and high heterogeneity, respectively.
Software (STATA) and the commands used for the meta-analysis (MIDAS and METANDI) are clearly stated together with the p-value < 0.05 considered for statistical significance. Figure 1 represents an excellent visual flow chart summary for the final six studies included in the meta-analysis, reporting cases in 3,974 women with 4,144 adnexal masses (excellent sample size).
Tables 1, 2, and 3; as well as Figures 2 through 6, clearly summarize and show the results of the meta-analysis, from study selection using PICOS criteria to HSROC curve for TVS diagnostic accuracy for hydrosalpinx. Overall, this meta-analysis study does a proficient use of statistical techniques and tools from sample selection to data processing and analysis.
The authors make a good point at stating that the main strength of their study is that it is the first meta-analysis addressing this topic. Additionally, they honestly recognize two weaknesses; 1. Even though the studies account for a good sample size of cases (>4,000), the number of studies analyzed is still low at only six. 2. The data used could be more updated since the studies analyzed are from 10 to 20 years ago and TVS technology performance has improved ever since.
The authors adequately conclude that their review and meta-analysis prove that TVS has good accuracy with the diagnosis of hydrosalpinx, with high specificity and sufficient sensitivity; and that it can be considered a reliable diagnostic technique of major importance in an era of assisted reproduction and IVF.
My only concern is the page numbering. Make sure the page numbering is correct.
Author Response
Dear Reviewer
Thanks for your comments. We appreciate them.
We have revised language and line numbering
Sincerely yours
Juan Luis Alcazar